# Impact of Insoluble Separation Layer Mechanical Properties on Disintegration and Dissolution Kinetics of Multilayer Tablets

**DOI:** 10.3390/pharmaceutics12060495

**Published:** 2020-05-29

**Authors:** Reiji Yokoyama, Go Kimura, Jörg Huwyler, Ken-ichi Hosoya, Maxim Puchkov

**Affiliations:** 1Department of Pharmaceutical Sciences, Division of Pharmaceutical Technology, University of Basel, Klingelbergstrasse 50, CH4056 Basel, Switzerland; reiji.yokoyama@shionogi.co.jp (R.Y.); joerg.huwyler@unibas.ch (J.H.); 2Formulation R&D Laboratory, CMC R&D Division, SHIONOGI & CO., LTD., Hyogo 660-0813, Japan; 3Department of Pharmaceutics, Graduate School of Medicine and Pharmaceutical Sciences, University of Toyama, Sugitani 2630, Toyama 930-0152, Japan; hosoyak@pha.u-toyama.ac.jp; 4Production Technology Department, Manufacturing Division, Shionogi Pharma Co., LTD., Hyogo 660-0813, Japan; go.kimura@shionogi.co.jp

**Keywords:** disintegration simulation, poorly water-soluble drug, mefenamic acid, acetylsalicylic acid, Noyes–Whitney equation, cellular automata

## Abstract

Dissolution and disintegration of solid dosage forms such as multiple-layer tablet with different active ingredients depend on formulation and properties used in the formulations, and it may sometimes result in counterintuitive release kinetics. In this manuscript, we investigate the behavior of combined acetylsalicylic acid and mefenamic acid bi- and triple-layer formulations. We show that the simulation model with a cellular automata predicted the impact of the inert layer between the different active ingredients on each drug release and provide a good agreement with the experimental results. Also, it is shown that the analysis based on the Noyes–Whitney equation in combination with a cellular automata-supported dissolution and disintegration numerical solutions explain the nature of the unexpected effects. We conclude that the proposed simulation approach is valuable to predict the influence of material attributes and process parameters on drug release from multicomponent and multiple-layer pharmaceutical tablets and to help us develop the drug product formulation.

## 1. Introduction

In the past decades, mostly fixed dose combination products (FDC) have been developed and introduced on the market [1]. In the development of FDC, the dosage form of the multi-layer tablets is selected since the multi-layer tablets have several advantages as compared to the monolithic conventional tablets. For example, one of the benefits of a multi-layer tablet geometries is a possibility to reduce chemical incompatibilities between active pharmaceutical ingredients (APIs) or between other formulation components by physical separation. Additional benefit of a multi-layer approach is a possibility to design dedicated release kinetics for each API in the dosage form [2,3] to help prolonging the product’s life cycle [4]. As an approach controlling and modifying the release rate of the API, the multilayer tablet technologies which combine the immediate release layers with the barrier or swellable/erodible properties was reported [5,6,7]. In those cases as well as in the case of multilayered pellet formulations, the drug release of the API was modified depending on the amount of hydrogel polymers such as hydroxy propyl methyl cellulose acetate succinate (HPMCAS) in the layer and the multi-layer design [8,9]. To understand the drug release mechanism from the experimental data set, kinetic mechanism studies with the mathematical fitting, for example, Higuchi model, Ritger–Peppas model and non-Fickian model, were reported [10,11]. However, it is challenging to predict the dissolution of the multi-layer tablets from the individual compressed layers of the same formulations as the layers in the multilayer tablets. As a reason, it is considered that there is a need for universal models reflecting the shape and other factors, e.g., components’ colligative properties, and for real tablets or layered pellets which may affect the dissolution [12], despite numerous successful attempts of modeling individual shape effects, especially models based on parabolic partial differential equations and extensions of Noyes–Whitney models with fractal geometry [7,13,14,15,16,17].Therefore, there is a need to link dissolution with disintegration especially for the mentioned multilayer formulations with nontrivial geometries.

Over the past decades, numerous simulation models were developed in the field of pharmaceutical tablets compaction, dissolution, and disintegration [18,19,20,21], for example, models of mechanical and dissolution behavior of particles and tablets [19,22]. Numerical simulation models based on finite-element method (FEM) treats the powder as a continuous material and is used for simulation of the drug release from hydrogel-based matrix tablets [23]. In case of a significantly higher degree of formulation heterogeneity, the continuous methods are less applicable, especially for those systems where point contacts between formulation components play a significant role, and the use of discrete methods is appropriate [24,25,26]. It was reported that swelling behavior and the drug release from hydrophilic polymer tablets with different tablet shapes, components, and drug loading can be simulated with the discrete-element method (DEM) [24,25,26,27]. Besides DEM, the modeling technique with the cellular automata algorithm was proposed [28,29], and it was used to simulate the drug release of pharmaceutical tablets [30,31]. Importantly, three-dimensional cellular algorithms allow simulation of formulations containing different types of components organized as a large number of discrete cubic mesh; the simplicity of the calculation with the cellular automata enable the simulation with many different types of components, as compared to DEM models [32]. For example, it was reported in our previous work that the simulation algorithms could simulate the disintegration and dissolution of the mefenamic acid (MA) tablets very close to real experimental data [33]. In the simulation, the molecular dynamics simulation was adopted for the calculation of the diffusion coefficients of the API molecule. Other applications of the three-dimensional cellular automata algorithm include simulation of the disintegration time of pharmaceutical tablets [21] and of the buoyancy and drug release simulations of gastroretentive floating tablets [34,35].

From the advantages of the three-dimensional cellular automata, the purpose of the study was to simulate the dissolution of the multi-layer tablets with the three-dimensional cellular automata and to elucidate the impact of the insoluble layer on the formulation performance of the multi-layer tablets.

## 2. Materials and Methods

### 2.1. Materials

Mefenamic acid (SIGMA, St. Louis, MO, USA) and acetylsalicylic acid (ASA) (Lot: W004649, Glatt GmbH, Binzen, Germany) were selected as a model compound to prepare rapidly disintegrating tablets. d-mannitol (Pearlitol 25C, Roquette, Lestrem, France) and microcrystalline cellulose (Avicel PH-101, FMC bioPolymer, Philadelphia, PA, USA) were used as diluents, and croscarmellose sodium (Ac-Di-Sol, FMC bioPolymer, Philadelphia, PA, USA) was used as a disintegrant. Hydroxypropyl cellulose (HPC SL, NIPPON SODA, Tokyo, Japan) was a binder, and magnesium stearate (Peter Greven GmbH & Co, Bad Münstereifel, Germany) as a lubricant was also selected. Polycaprolactone (1639, Abifor, Zurich, Switzerland) was used for the inert 2nd layer of the triple-layer tablet. Cetyltrimethylammonium bromide (CTAB) (Merck, Kenilworth, NJ, USA) was used as a surfactant for dissolution tests.

### 2.2. Methods

#### 2.2.1. Preparation of Tablets

In this study, two single entities of ASA tablets and MA tablets were prepared. The formulation of the ASA tablets used for this study was presented in Table 1. For preparing the ASA tablet, ASA was granulated with the weighed excipients (i.e., D-mannitol, microcrystalline cellulose, croscarmellose sodium, and hydroxypropyl cellulose) in a high-shear mixer (MYCROMIX, OYSTAR Hüttlin, Schopfheim, Germany). The powders were mixed for 1 min at an impeller speed of 250 rpm prior to the granulation process. Next, the granulation process was carried out at 250-rpm impeller speed/2000-rpm chopper speed. Hydroxypropyl cellulose aqueous solution (10% *w*/*w*) was added at a spray rate of approximately 5 g/min. After completion of the binder solution, water was added at the same spray rate to flush the line. The process was continued for 1 min. The obtained wet granules were dried and milled using a screen mill (Fitz mill model L1A, Fitz Patrick, Waterloo, ON, Canada). Afterward, the milled granules were weighed and mixed with croscarmellose sodium and magnesium stearate. The ASA tablets were prepared using a compaction simulator (StylOne, Medel pharma, Beynost, France) equipped with a tooling of 21.5 × 10-mm oval punch set. The setting value of the dwell times for pre-compression and main compression were 25 ms. The compaction parameters are presented in Table 1. With respect to the MA tablets, the MA granulate and the tablet was prepared according to previous work [33].

The bi-layer tablet consists of MA and ASA layers, and the triple-layer tablet consists of MA, polycaprolactone, and ASA layers. The formulations of the bi-layer tablet and triple-layer tablet are shown in Table 2. The 1st layer, middle layer, and 3rd layer of the triple-layer tablet were ASA, polycaprolactone, and MA, respectively. The same ASA granulate and the MA granulate, which was used for each single layer tablet, were used in the multilayer tablets and compacted with the same tooling, which was used for single API tablets. The dwell times for pre-compression and main compression were set to 25 ms. The compaction parameters are presented in Table 2. 

#### 2.2.2. Determination of Tablet Porosity, Effective Compressive Stress, and Surface Area

Mean tablet weight was evaluated with an electronic balance (AX204 Delta Range, Mettler Toledo, Greifensee, Switzerland). In addition, tablet diameter and tablet thickness were evaluated with a digital caliper (CD-15CPX, Mitutoyo, Kanagawa, Japan). All obtained values of tablet weight, diameter, and thickness were within 1% deviation. True densities of all raw materials were evaluated using helium pycnometry (AccuPyc 1330, Micrometrics, Norcross, GA, USA). The values of the ASA and polycaprolactone are 1.3928 g/cm^3^ and 1.1600 g/cm^3^, respectively. The values for other ingredients were reported previously [33]. 

The true density of each layer in the tablets was calculated according to Equation (1).
(1)$$\rho_{layer}=\frac{1}{\sum_{i=1}^{n}\frac{X_{i}}{\rho_{i}}}$$
where *ρ_layer_* and *ρ_i_* are the true densities (g/cm^3^) of the layer and each raw material in the layer, respectively, and *X_i_* is the weight fraction of each component.

The porosity ε of the layers was determined according to Equation (2).
(2)$$\varepsilon =1-\frac{m}{V\rho_{layer}},$$
where *V* is the layer volume calculated from the experimental layer thickness and F-CAD software and where m is the weight of the layer.

The horizontal cross-sectional area and the surface area of the tablets were calculated using the computer-aided design software T-CAD (T-CAD, Tokyo, Japan), and the effective compressive stress *P_comp_* (MPa) was calculated according to Equation (3).
(3)$$P_{comp}=\frac{F_{comp}\times 1000}{S_{cross\;section}}$$
where *S_cross section_* (mm^2^) is the horizontal cross section area of the tablets and *F_comp_* (kN) is the resultant compressive force in the compression of the tablets. 

#### 2.2.3. Disintegration Test

The disintegration times were measured using a disintegration tester (Sotax DT3, Sotax AG, Allschwil, Switzerland), according to the United States Pharmacopeia (USP) 24 method. Tests were carried out in 900 mL of 50 mM sodium phosphate buffer (pH 6.8) containing 1% CTAB at 37 °C ± 0.5 °C (*n* = 3). All tests were done in triplicate using six tablets for each test.

#### 2.2.4. Dissolution Test

Dissolution tests of the ASA tablets, the bilayer tablets, and the triple layer tablets were carried out using the USP dissolution apparatus II with a paddle rotation of 75 rpm (*n* = 6) (AT7smart, Sotax, Allschwil, Switzerland). The dissolution media and its volume is 900 mL of 50 mM sodium phosphate buffer (pH 6.8) containing 1% CTAB at 37 °C ± 0.5 °C. Drug rate were analyzed by using an HPLC system (Agilent 1100 LC, Santa Clara, CA, United States) equipped with a reverse phase T3 Atlantis C18 column (3 um, 3.0 × 20 mm, Waters, MA, USA) every 5 min. The wavelength of the UV detector was set to 230 nm and 237 nm for MA and ASA, respectively. The column temperature was maintained at 40 °C. The mobile phase used for MA and ASA was acetonitrile/10 mM ammonium formate buffer solution (pH 3.5). The bilayer tablet and triple layer tablet were inserted into the dissolution vessels to ensure that the MA layer is the upper side in the bottom of the vessels. (to avoid the variation of dissolution due to difference in the layer position)

#### 2.2.5. Water Sorption Test

Liquid sorption was measured for MA and ASA granulates in order to compare liquid imbibition rates of both layer components. Measurements were carried out with Krüss Tensiometer K100 (Krüss GmbH, Hamburg, Germany), and data were analyzed with Laboratory Desktop, Ver. 3.2.2.3068 (Krüss GmbH, Germany). The weight of the granulates used for experiments was approx. 700 mg; the inner diameter of the measurement cell was 11.33 mm. The liquid medium was distilled water. Time for liquid uptake experiment was taken enough to reach a plateau region, which indicates no significant further weight increase. The slope and capillary constants were calculated with the Laboratory Desktop software after manual selection of linear sections on the squared mass vs. time plots. Every sorption experiment was carried out in triplicates.

### 2.3. Simulation of Drug Release with Cellular Automata

The software package F-CAD v.2.0 was used for the simulations of the drug release of ASA tablets, the bilayer tablets, and the triple-layer tablets. For the simulation of the experimental tablet, the virtual tablets created in the software, for which geometrical information (i.e., size, shape, and layer order) is identical to the experimental tablets (i.e., 21.5 × 10-mm oval tablet), were generated. The virtual tablet was discretized into a cubic grid using a voxel side length of 74 μm (with 293^3^ elements). In our previous work [33], the molecular dynamics approach was adopted to calculate the simulation constants of MA particle, which expresses the dissolution rate in the simulation model with a cellular automata; thus, the same method of the molecular dynamics approach was adopted for the calculation of ASA simulation constants as well.

#### Comparison of Drug Release Pattern between Experimental and Simulated Profiles

To calculate the similarity factor (*f*_2_) of the drug release between experimental data and simulated data, Equation (4) was used [36].
(4)$$f_{2}=50\times \log\left\{{\left[1+\frac{1}{n}\sum_{i=1}^{n}{\left(R_{t}-T_{t}\right)}^{2}\right]}^{-0.5}\times 100\right\}$$
where *n* is the number of time points, *R_t_* is the dissolution rate of the experimental tablet at time *t*, and *T_t_* is the dissolution rate of the simulated tablet at time *t*. A similarity factor (*f*_2_) greater than 50 indicates a close correlation between simulated and experimental data.

### 2.4. Fitting of Noyes–Whitney Dissolution Model with Experimental Data

Fitting of the experimental dissolution results were made with Mathematica 11.0 (Wolfram Research Inc., Champaign, IL, USA). The screenshot is introduced in Appendix A. The source code is available in the supplement. The data were fitted with the Noyes–Whitney equation:(5)$$\frac{dC}{dt}=\frac{DS}{h}\left(C_{s}-C\right),$$
where *C* is the fraction of released drug (%, *w*/*v*), *Cs* is the drug solubility (%, *w*/*v*), *D* is the diffusivity coefficient (cm^2^/s), *h* is the length of the unstirred layer (cm), and *S* is the specific surface of the sample (cm^2^/g). 

Unlike the dissolution and disintegration simulations with three-dimensional cellular automata, the model of Noyes–Whitney does not consider the spatial differentials and material fluxes, which mainly govern the dynamics of dissolution and disintegration of the tablets. The main use and purpose of the Noyes–Whitney equation in this manuscript was to obtain a single coefficient fitting model for comparative analysis of the experimental data. Specific surface was the sole fitting coefficient selected for this purpose. Fittings were made for release data of ASA layer for both triple- and bi-layer tablets. The obtained values for S were interpreted as surface available for dissolution contact in both tablet geometries. 

## 3. Results

### 3.1. In Vitro Evaluation of Drug Release

The properties of the experimental tablets and their compaction condition are summarized in Table 3. Also, with respect to the effective surface area contributing the disintegration and dissolution of the tablets in the dissolution tests, it is considered in the triple-layer tablet that the effective surface area for MA layer and the ASA layer decreased by the contacting polycaprolactone layer, which does not penetrate the water into these adjacent layers. Based on the theory, the effective surface area was calculated and given in Table 4. 

In vitro drug release testing of the MA tablets, the ASA tablets, the bilayer tablets, and the triple-layer tablets were carried out, and the results of MA and ASA are shown in Figure 1 and Figure 2, respectively.

The obtained release profiles for MA in Figure 1 are identical for all study cases, i.e., single tablet, bilayer tablet, and triple layer tablet. An unexpectedly different result is seen for ASA (Figure 2), with significantly slower release rates in the case of a triple-layer tablet formulation. As shown in Table 3, the disintegration time of the bilayer tablet and the triple-layer tablet are longer than that of the single component tablets, corresponding to the rank-order of the in vitro dissolution data. 

Results of the fitting of the experimental data with differential form of the Noyes–Whitney equation yields 1.8 times reduction in contact surface for ASA in triple-layered tablet (R^2^ = 0.99).

The results of the liquid uptake measurements for the granulates used to compact the layer of bi- and triple-layered tablet are show in Table 5. The total amount of the adsorbed liquid for both granulates was not significantly different. On the other hand, the rates of the liquid imbibition, i.e., a slope and the capillary constant, show differences by factors of approx. 6 and 4, respectively.

### 3.2. Comparison of in Silico Drug Release Profiles between Bilayer Tablet and Triple-Layer Tablet

The in silico drug release of MA and ASA from the bilayer tablets and the triple layer tablets, which was simulated with cellular automata-based numeric models, are shown in Figure 3, respectively. Also, the comparison between in silico and in vitro release profiles from the bilayer tablets and triple-layer tablets are shown in Figure 4 and Figure 5, respectively.

An attempt to simulate the dissolution profiles of individual layers and the tablet as a whole resulted in the simulated curves of high similarity to the experimentally obtained data, therefore, suggesting similarity between real physical processes and their simulated counterparts. In Table 6, the corresponding *f*_2_ factors are given as a measure of similarity between simulated and experimental release profiles of MA and ASA bi- and triple-layered tablets.

## 4. Discussion

The experimental results shown in Figure 1 and Figure 2 were not expected in the case of ASA triple-layered tablet. The release profiles suggest significant reduction in the release rate for highly soluble compound as compared to the unchanged release rate of the low-soluble MA component. The change is only observed if an inert layer of the polycaprolactone is introduced between layers, i.e., in the statistical analysis (one-way ANOVA) of dissolution rates at 10 min, 15 min, and 30 min, *p*-value < 0.05 is only observed when comparing the dissolution rates between ASA tablets and triple-layer tablets. It is important to say that this effect cannot be simply explained by a geometrical contact surface reduction for ASA layer through MA or the inert separation layer as, in such a case, a similar effect would be observed for bi-layer tablets, i.e., tablets without separation component. On the contrary, the same behavior cannot be repeated for identically layered tablet composition without an intermediate separation. The results of fitting analysis suggest that ASA layer in a triple-layered tablet has reduced contact surface (by 1.8 times, according to Noyes–Whitney model-based analysis) exposed towards the dissolution medium. At the same time, the ASA layer is a highly soluble formulation and should have shown little or no dependency on the drug release from the MA layer. The results of the liquid uptake measurements corroborate the latter assumption. As shown in Table 5, the rate of liquid uptake for ASA layer is approx. 6 times faster than the capillary uplift registered for MA granules. The analysis of the simulated and experimental data of a bi-layer tablets, i.e., the tablets without inert separation layer, confirms that the upper layer with MA formulation is disintegrated faster due to its reduced thickness as compared to ASA layer and, hence, introduces no more obstacle for a liquid front to ingress into ASA layer. The Polycaprolactone (PCL) material apparently blocks this action for triple-layered tablets, as it remains attached to the surface of the ASA layer long after complete disintegration of the MA layer. The sensitivity of the ASA formulation to the geometry of the tablet was not expected, as it is normally not observed for fast disintegrating immediate release tablet formulations with highly soluble active substances. However, neither the three-dimensional cellular algorithms simulation of the drug release nor fitting of experimental data with the Noyes–Whitney model suggest an alternative explanation of the role of the inert layer. Similar findings reported in Reference [37] were also explained by different surface-to-volume ratios for single- and multi-layer formulations. The disintegration mechanism of ASA layer is different to the one of MA layer and can be best characterized as being erosion-driven. The difference in the disintegration mechanisms for almost identical formulations, i.e., not taking into account the geometrical differences, can be best explained by different diffusivities and aqueous solubilities of the active substances. Interestingly, the visual observation of the tablets’ behavior during dissolution suggests rather limited liquid imbibition despite sufficient porosity; liquid transport in the porous meshwork is, apparently, not intensive enough to cause an immediate disintegration; therefore, an erosion-driven disintegration occurs. An effect of porous transport passivation may be caused by an increased viscosity of intra-porous liquid due to dissolved tablet components, such as ASA and mannitol. On the other hand, porosity of the ASA layer was lower as compared to MA (Table 3), which would allow only limited liquid imbibition into tablet body. In order to increase liquid sorption rate and, therefore, to induce disintegration and consequently dissolution speed, the porosity of the pharmaceutical compacts should be greater than the critical value, known as a percolation threshold [38]. However, the higher porosities of tablets lead to reduced tensile strength of the compact; therefore, such an approach to increase liquid diffusivity should be taken with care.

Another important aspect which was identified in this study is suitability of the two dissolution models. The Noyes–Whitney model, which accounts for basic characteristics of dissolving solid, e.g., diffusion and solubility parameters, can be applied for general description of the dissolution behavior under the assumption that the dissolving solid is a homogeneous solid. Significantly better modeling results can be obtained with models based on numerical methods, such as cellular automata method, used in this study. In the left side of Figure 5, the experimental release data of ASA are modelled with cellular automata (solid line). It is possible to spot a slightly sigmoidal shape of the modelled curve generated by the simulation. The models based solely on the Noyes–Whitney kinetic will not produce similar curves without significant extension of the model equations and introduction of further assumptions or simplifications.

The results of the study suggest that, for practical design of the multiple layer tablets, the intermediate layer must be designed as either a soluble or fast disintegrating component to avoid reduction of the dissolution rate, unless otherwise desired.

Inert layer material selection is not a trivial task, especially if this material must be compressed into fast disintegrating layers during tablet production. The layer must maintain separation integrity in order to effectively separate the active ingredients for enhanced stability of the final medicinal product.

## 5. Conclusions

In this study, the drug release from triple-layer multicomponent tablet formulation of mefenamic and acetylsalicylic acids was evaluated in vitro and in silico. Numerical simulations were applied to reveal the role of inert separation layer on the changing release behavior of the bi- and tri-layer tablets. For in silico simulation, the model with the three-dimensional cellular automata algorithm showed good agreements with the experimental dissolution data and predicted the impact of the insoluble layer on the dissolution of the APIs in both layers. The unmodified Noyes–Whitney dissolution model does not take disintegration into account and, therefore, was used only as a fitting model for comparative purposes. It is concluded that the simulation based on numerical methods, such as cellular automata, can help to understand the influence of different materials and their attributes, geometry, and process parameters on drug release from multicomponent and multiple-layer pharmaceutical tablet and can aid in rapid drug product formulation.

The results of this study identify the impact of the insoluble layer on the formulation performance of multiple-layer tablets. Introduction of the insoluble and inert layers may significantly reduce the dissolution speed due to reduced surface of both layers. This effect is present even if the bi-layer tablets composed of the same materials do not show dissolution rate reduction. This often-undesired effect will disappear if the separation layer can be composed of either fast soluble or fast disintegrating material. 

## Figures and Tables

**Figure 1 pharmaceutics-12-00495-f001:**
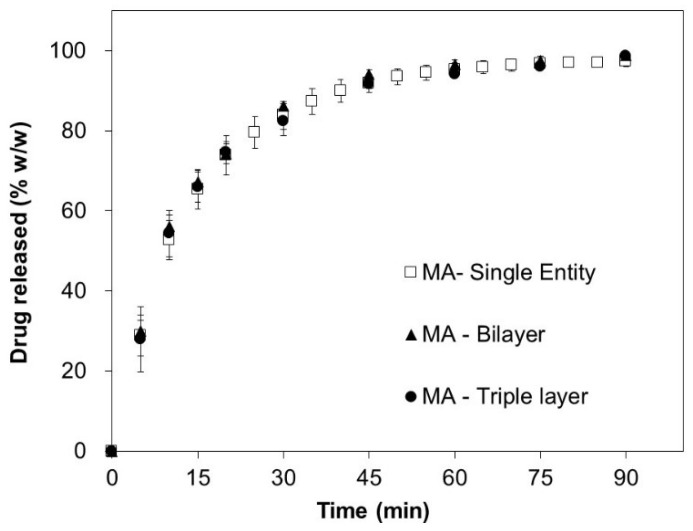
Experimental drug release of mefenamic acid (MA).

**Figure 2 pharmaceutics-12-00495-f002:**
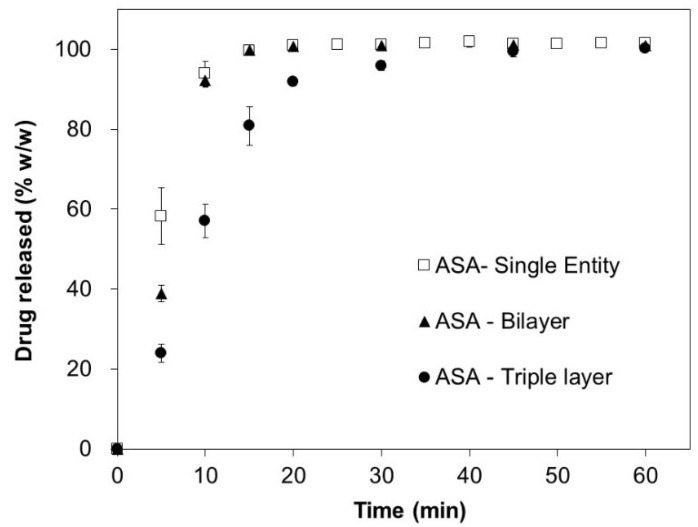
Experimental drug release of acetylsalicylic acid (ASA).

**Figure 3 pharmaceutics-12-00495-f003:**
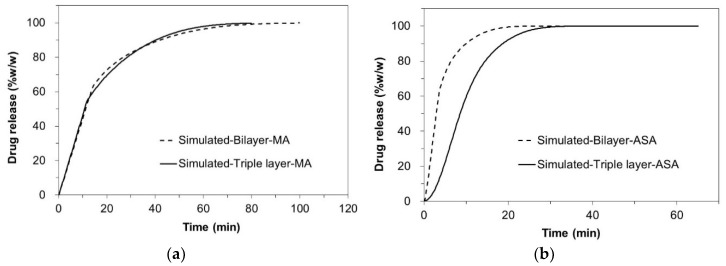
Comparison of in silico release profiles between the bilayer tablets (dotted lines) and the triple-layer tablets (solid lines) (**a**) MA; (**b**): ASA.

**Figure 4 pharmaceutics-12-00495-f004:**
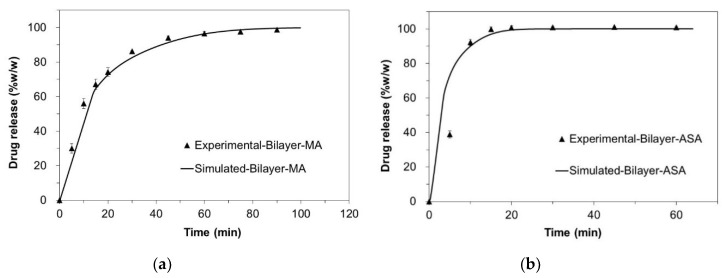
Comparison between in silico (solid lines) and in vitro (filled triangle) release profiles from the bilayer tablets (**a**) MA; (**b**) ASA.

**Figure 5 pharmaceutics-12-00495-f005:**
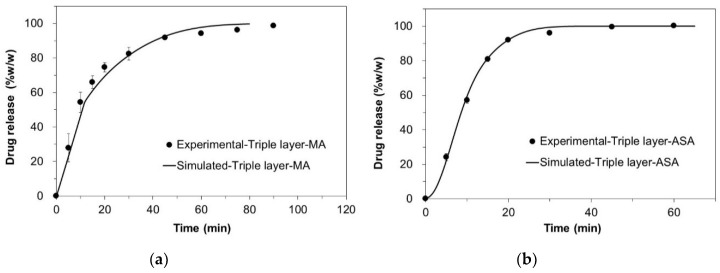
Comparison between in silico (solid lines) and in vitro (filled circles) release profiles from the triple layer tablets (**a**) MA; (**b**) ASA.

**Table 1 pharmaceutics-12-00495-t001:** Formulation compositions and tablet compaction parameters. ASA: acetylsalicylic acid.

Formulation Composition	ASA Tablet	
	mg	%, *w*/*w*
Granule composition		
ASA	500.0	50.0
d-mannitol	330.0	33.0
Microcrystalline cellulose	100.0	10.0
Croscarmellose sodium	20.0	2.0
Hydroxypropyl cellulose	30.0	3.0
Granulate	980.0	98.0
External phase composition		
Croscarmellose sodium	10.0	1.0
Magnesium stearate	10.0	1.0
Tablet weight	1000.0	-
Tablet Parameters (*N* = 9)		
Tablet porosity (%, *v*/*v*)	7.1	
Compressive force (kN)	26	

**Table 2 pharmaceutics-12-00495-t002:** Formulation compositions and tablet compaction parameters. MA: mefenamic acid.

Formulation Composition	Bi-Layer Tablet		Triple-Layer Tablet	
	mg	%, *w*/*w*	mg	%, *w*/*w*
Granule composition				
ASA	500.0	50.0	500.0	50.0
d-mannitol	330.0	33.0	330.0	33.0
Microcrystalline cellulose	100.0	10.0	100.0	10.0
Croscarmellose sodium	20.0	2.0	20.0	2.0
Hydroxypropyl cellulose	30.0	3.0	30.0	3.0
Granulate	980.0	98.0	980.0	98.0
External phase composition				
Croscarmellose sodium	10.0	1.0	10.0	1.0
Magnesium stearate	10.0	1.0	10.0	1.0
ASA layer weight	1000.0	-	1000.0	-
Separation layer Polycaprolactone (PCL)	-	-	300.0	-
Granule composition				
MA	250.0	50.0	250.0	50.0
d-mannitol	165.0	33.0	165.0	33.0
Microcrystalline cellulose	50.0	10.0	50.0	10.0
Croscarmellose sodium	10.0	2.0	10.0	2.0
Hydroxypropyl cellulose	15.0	3.0	15.0	3.0
Granulate	490.0	98.0	490.0	98.0
External phase composition				
Croscarmellose sodium	5.0	1.0	5.0	1.0
Magnesium stearate	5.0	1.0	5.0	1.0
MA layer weight	500.0	-	500.0	-
Tablet weight	1500.0	-	1800.0	-
Tablet porosity (%, *v*/*v*)	7.4 (ASA layer) 11.1 (MA layer)		7.4 (ASA layer) 11.1 (MA layer)	
Compressive force (kN)	21		21	

**Table 3 pharmaceutics-12-00495-t003:** Tablet properties and compaction conditions.

Formulation	Effective Compressive Stress (MPa)	Tablet Weight (mg) (*n* = 6)	Tablet Thickness (mm) (*n* = 6)	Porosity-MA (%, *v*/*v*)	Porosity-ASA (%, *v*/*v*)	Disintegration Time (s)
MA tablet	99	500.3 ± 1.1	4.22 ± 0.02	13.7	NA	160 ± 4
ASA tablet	155	1001.0 ± 2.4	6.80 ± 0.02	NA	7.1	81 ± 6
Bilayer tablet	127	1502.7 ± 1.4	9.17 ± 0.01	7.4	11.1	228 ± 18
Triple layer tablet	126	1501.1 ± 0.8	10.75 ± 0.02	7.4	11.1	289 ± 24

**Table 4 pharmaceutics-12-00495-t004:** Comparison of tablets’ surface area and areas of individual layers.

Item	MA Tablet	ASA Tablet	Bilayer Tablet		Triple-Layer Tablet	
			MA Layer	ASA Layer	MA Layer	ASA Layer
Surface area (mm^2^)	350.2	484.4	289.3	318.4	289.3	318.4
% of Control	MA Control	ASA Control	83%	66%	83%	66%

**Table 5 pharmaceutics-12-00495-t005:** Results of the liquid uptake measurements for MA and ASA granulates (n = 3, error range is given as standard deviation).

	Maximum Liquid Uptake, % (w/w)	Slope, g^2^/s	Capillary Constant × 10^−6^
MA	78 ± 3.5	0.013 ± 4.5 × 10^−3^	1.81 ± 0.63
ASA	83 ± 5.7	0.064 ± 1.4 × 10^−2^	8.84 ± 1.94

**Table 6 pharmaceutics-12-00495-t006:** Summary of similarity factors (*f*_2_) between in vitro and in silico drug release profiles.

Tablet		MA	ASA
Similarity factor (*f*_2_)	Bilayer tablets	57	86
	Triple layer tablets	75	87

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
