# Peer review of "Impact of Insoluble Separation Layer Mechanical Properties on Disintegration and Dissolution Kinetics of Multilayer Tablets"

_pharmaceutics, 2020, doi:10.3390/pharmaceutics12060495_

Round 1
Reviewer 1 Report
The current manuscript focuses on modeling the release behavior from bi-layered and tri-layered tablets. The authors did a good job in explaining the knowledge gap, compact set of experiments and discussion. The article in its current form pending some minor English language checks can be deemed acceptable to me.
Author Response
We are thankful for the time spend by the reviewers to help improve this manuscript. All comments are well taken.
Reviewer-1 Comments
The current manuscript focuses on modeling the release behavior from bi-layered and tri-layered tablets. The authors did a good job in explaining the knowledge gap, compact set of experiments and discussion. The article in its current form pending some minor English language checks can be deemed acceptable to me.
We are thankful for time and effort spend by reviewer to help us improve this manuscript. The grammar and style of the manuscript were improved as suggested.
Reviewer 2 Report
The manuscript entitled “Impact of insoluble separation layer mechanical properties on disintegration and dissolution kinetics of multilayer tablets” study the drug release behavior through experiment and simulation. However, The material, content and discussion of this article are relatively simple, but also lack of more systematic data and in-depth explanation. Therefore, I recommend rejecting the manuscript for publication in Pharmaceutics. Several specific comments are listed as follow:
- Mefenamic acid and acetylsalicylic acid appear for the first time in full terms, and it is suggested that abbreviations be used later, which agree with they shown in the figures.
- Figure 6: The experimental in vitro release datafrom the triple layer tablets of ASA looks different from that in figure 3, please check. Whether Figure 6 should be "Fitting of Noyes-Whitney dissolution model with experimental data"?
- “Fitting of Noyes-Whitney dissolution model with experimental data”: The formulation, drugs distribution, interaction between drugs and carriers, carriers porosity, carriers thickness and other factors affect the dissolution and diffusion process of drugs, and then affect drug release, whether formula 4 is too simple? Besides, there is lack of related detailed results and analysis.
- This article is lack of making detailed comparison with other previous systems in introduction and putting forward the main innovation in results.
Author Response
Rebuttal letter
We are thankful for the time spend by the reviewers to help improve this manuscript. All comments are well taken. Our answers to the original questions are marked in blue and the changes in the manuscript are marked green.
Reviewer-2 Comments
- Mefenamic acid and acetylsalicylic acid appear for the first time in full terms, and it is suggested that abbreviations be used later, which agree with they shown in the figures.
This comment is well accepted, all corresponding abbreviations were introduced into the text.
- Figure 6: The experimental in vitro release data from the triple layer tablets of ASA looks different from that in figure 3, please check. Whether Figure 6 should be "Fitting of Noyes-Whitney dissolution model with experimental data"?
We are thankful for this valuable comment, we can confirm that ASA release from triple layer tablet show in Figure 2 (Figure 3 in the original manuscript) and in Figure 5 (Figure 6 in the original manuscript) are the same. However, it might be difficult to see on the charts, therefore the following sentence was added at the line 200 in the original manuscript “which was simulated with cellular automata”. The solid lines on the Figure 5 correspond to cellular automata-based dissolution simulation and not the result of the Noyes-Whitney model fitting.
- “Fitting of Noyes-Whitney dissolution model with experimental data”: The formulation, drugs distribution, interaction between drugs and carriers, carriers porosity, carriers thickness and other factors affect the dissolution and diffusion process of drugs, and then affect drug release, whether formula 4 is too simple? Besides, there is lack of related detailed results and analysis.
This is an important comment and it is well taken. Indeed, the Noyes-Whitney model (former eq. 4, now eq. 5) is valid mostly for single crystalline component dissolution under idealized conditions. This model itself is not considering neither porosity nor influence of other components, such as disintegrates or other APIs and was used only as a physically relevant fitting model for comparative reasons, where contact surface has been selected as a benchmark parameter. The detailed analysis of the numerical calculation model of the tablet formulation using discreet linear approximations of the integral-algebraic-differential models defined as cellular automata rules were addressed in our previous paper [30] in sufficient details, therefore we only reference it here. The results of the cellular automata dissolution modeling of the given formulations, including the geometry aspects are shown in Figure 5. Numerical simulations were carried out with 1 second time resolution and show very good agreement with experimental data. Simple fitting of the unmodified Noyes-Whitney model yields sufficiently less accurate fitting of the experimental data as commented by Reviewer 2.
- This article is lack of making detailed comparison with other previous systems in introduction and putting forward the main innovation in results.
We are thankful for these suggestions to improve the presentation of our manuscript. We have tried to improve the introduction and have introduced additional references as suggested by Reviewer 3. We have improved the Introduction and Conclusion sections to emphasize the counterintuitive influence of the separation layer on the performance of the triple layer formulations as compared to single layer tablets.
Reviewer 3 Report
Reviewer comments for the manuscript: Impact of insoluble separation layer mechanical properties on disintegration and dissolution kinetics of multilayer tablets
The authors have investigated the drug release behavior of different tablets formulations (bi and tri-layer tablets) composed by acetylsalicylic acid and mefenamic acid. They have applied some integral (Noyes-Whitney equation) and modeling solutions (FEM, DEM and cellular-automata methods) in order to elucidate and predict the effect of different excipients in the drug release mechanism and in the control release layer formation for these tablets. As a conclusion, authors state that there is an insoluble layer formation that affects the performance of multiple-layer tablets reducing the dissolution speed.
Comment 1: The abstract must be re-structured in order to provide a more concise idea related with the research addressed in the manuscript. Also, there are some grammar mistakes.
Comment 2: Line 29, there is a grammar mistake, I understand: “In decades, many fixed dose combination products (FDC) have been developed”
Comment 3: Lines 33 to 34, the presented idea in these lines are not clear. Also, the references [2] and [3] that were cited are not related with the supported idea.
Comment 4: Lines 43 to 45, The authors do not stablish a clear relationship between shape features of the tablets with the disintegration studies in order to obtain mathematical models for elucidate the drug release from multi-layer tablets
Comment 5: Line 47 Is this a typing mistake? “the model was applied to model…”
Comment 6: Line 59, check if there is a typing mistake
Comment 7: In lines 49 to 64, the authors presented some references about several modeling or simulation tools (FEM, DEM, Cellular-automata, molecular dynamics, etc.) which could be used for the elucidation of drug release from solid polymeric matrix. Nevertheless, the authors do not indicate which models will be using.
Comment 8: Lines 82 to 91, it is not clear the method to tablet preparation and the tablet differences.
Comment 9: Table 1 and 2, The weight for each tablet is not clear. In table 1, there was indicated that total tablet weight is 500 mg, but the authors have indicated that granulate weight was 980mg; the same in table 2. The tables must be explained in a better way
Comment 10: Lines 128 to 133. Is this dissolution test method a USP monographic method? It looks is not, why the author have used this?
Comment 11: Lines 146 to 148, Why the authors used the same method of the molecular dynamics approach but in the introduction section, their focus was to FEM and DEM models?
Comment 12: Table 3, effective compressive stress calculation method is not indicated in materials and methods section
Comment 13: Figure 2 and 3, error bars, legend and resolution of the figures must be improved. This figures are not clear and they do not help the understanding of the discussion.It is recommended the use of Graphpad prism software to improve the figures and the legends
Comment 14: Figures 5 and 6 are not clear. They have a lot of dots and other signs that cause confusion in the readers
Comment 15: Lines 221 to 225; These lines lack of scientific discussion about the release profiles, the authors just indicated that there is a significant reduction in release, but they do not present statistical arguments for that. In addition, what about the diffusion process for the active principles in the aqueous media? this aspect could improve the discussion due to the inert layer provide a boundary condition for drug diffusion to medium. The used software could be used to analyze some parameters about the Fick´s Diffusion Law.
Comment 16: Lines 228 to 230. A detailed revision of limitation to drug diffusion and data analysis must be done. The erosion and swelling process for the inert layers must be addressed. It is recommended the following references to check:
Nikolett Kállai, Oliver Luhn, et.al. Evaluation of Drug Release From Coated Pellets Based on Isomalt, Sugar, and Microcrystalline Cellulose Inert Cores. AAPS PharmSciTech. 2010 Mar; 11(1): 383–391
Kállai-Szabó N, et.al. Comparative dissolution study of drug and inert isomalt based core material from layered pellets. J Pharm Biomed Anal. 2014 Sep;98:339-44. doi: 10.1016/j.jpba.2014.06.005. Epub 2014 Jun 10.
Wei HE, et.al. Design and in Vitro/in Vivo Evaluation of Multi-layer Film Coated Pellets for Omeprazole. Chem. Pharm. Bull. 57(2) 122—128 (2009).
Kyu-Mok Hwanga, et.al. Swellable and porous bilayer tablet for gastroretentive drug delivery: Preparation and in vitro-in vivo evaluation. International Journal of Pharmaceutics, Volume 572, 15 December 2019, 118783
Comment 17: In lines 231 to 233. There is not a clear relationship between the geometry of the tablet and the ASA tablets disintegration rate. This section is lacking order and structure.
Comment 18: In lines 233 to 234
Comment 19: The authors talk about the difference in the experimental results, compared to the Noyes and Whitney model; however, the results are not adequately discussed, since the assumptions considered for the application of the Noyes-Withney model are not indicated
Comment 20: In lines 235 to 238 the authors indicated some aspects about the liquid imbibition; nevertheless, they did not any experiments where they addressed the liquid imbibition characterization, this evaluation does not appear in materials and methods section, it is just a speculation
Comment 21: In lines 239 to 245, the authors gave a discussion about disintegration and it relationship with dissolution profiles, also they have implicated that the inclusion of calcium carbonate to the formulation could improve a fast disintegration but they do not conducted any experiment which supported these types of affirmations. The last discussion is not supported enough
Comment 22: In the concluding section, the authors agree on the impact of the inert layer on the tablets. However, they had indicated that one of the objectives of the work was to in-silico modelling to simulate the drug dissolution profiles, even much of the introduction was on the FEM, DEM, cellular-automata simulation models, among others. However, the authors do not discuss or conclude on the simulation profiles and therefore the work lacks clear objectives and a logical and analytical order.
Author Response
Rebuttal letter
We are thankful for the time spend by the reviewers to help improve this manuscript. All comments are well taken. Our answers to the original questions are marked in blue and the changes in the manuscript are marked green.
Reviewer-3 Comments
The authors have investigated the drug release behavior of different tablets formulations (bi and tri-layer tablets) composed by acetylsalicylic acid and mefenamic acid. They have applied some integral (Noyes-Whitney equation) and modeling solutions (FEM, DEM and cellular-automata methods) in order to elucidate and predict the effect of different excipients in the drug release mechanism and in the control release layer formation for these tablets. As a conclusion, authors state that there is an insoluble layer formation that affects the performance of multiple-layer tablets reducing the dissolution speed.
Thank you for careful analysis of this study. We would like to pinpoint that our main aim was to investigate the effect of an insoluble layer introduction on the dissolution performance in constrained geometries. We have applied fittings of the Noyes-Whitney dissolution model and cellular-automata numerical model of the dissolution simulation of heterogeneous systems, such as triple layered multicomponent combined formulations.
Comment 1: The abstract must be re-structured in order to provide a more concise idea related with the research addressed in the manuscript. Also, there are some grammar mistakes.
Thank you for providing this suggestion. The abstract was adapted to better reflect the aim and the main results of the manuscript.
Dissolution and disintegration of solid dosage forms such as multiple-layer tablet with different active ingredients depend on formulation and properties used in the formulations, and it may sometimes result in counter-intuitive release kinetics. In this manuscript we investigate the behavior of combined acetylsalicylic acid and mefenamic acid bi- and triple-layer formulations. We show that the simulation model with a cellular automata predicted the impact of the inert layer between the different active ingredients on each drug release and provide a good agreement with the experimental results. Also, it is shown that the analysis based on the Noyes-Whitney equation in combination with a cellular automata-supported dissolution and disintegration numerical solutions explain the nature of the unexpected effects. We conclude that the proposed simulation approach is valuable to predict the influence of material attributes and process parameters on drug release from multicomponent and multiple-layer pharmaceutical tablets and help us develop the drug product formulation.
Comment 2: Line 29, there is a grammar mistake, I understand: “In decades, many fixed dose combination products (FDC) have been developed”
Thank you for the careful reviewing. The sentence in Line 29 was revised.
In the past decades mostly fixed dose combination products (FDC) have been developed and introduced on the market
Comment 3: Lines 33 to 34, the presented idea in these lines are not clear. Also, the references [2] and [3] that were cited are not related with the supported idea.
Thank you for the careful reviewing. The sentence of Line 33 to 34 was revised, and the appropriate references related to the idea were introduced.
In the development of the FDC, the dosage form of the multi-layer tablets is selected since the multi-layer tablets have several advantages as compared to the monolithic conventional tablets. For example, one of the benefits of a multi-layer tablet geometries is a possibility to reduce chemical incompatibilities between active pharmaceutical ingredients (APIs) or between other formulation components by physical separation. Additional benefit of a multi-layer approach is a possibility to design dedicated release kinetics for each API in the dosage form [2,3] to help prolonging the product’s life cycle
Comment 4: Lines 43 to 45, The authors do not establish a clear relationship between shape features of the tablets with the disintegration studies in order to obtain mathematical models for elucidate the drug release from multi-layer tablets
This comment is well accepted. We agree with the Reviewer that tablet disintegration is a complex and heterogeneous process, which depend on the number of factors, including tablets’ geometry. This important statement very well supported in our previous paper [30] where were applying the cellular automata numerical model to simulate the drug release from disintegrating formulations. In this manuscript, in order to reflect on this important physical phenomenon, we were able to introduce the missing experimental data on disintegration of single layer, bi-layer and triple-layer tablets. The methods section was updated to include the instruments and a method of the disintegration time determination.
The disintegration times were measured using a disintegration tester (Sotax DT3, Sotax AG, Allschwil, Switzerland), according to the United States Pharmacopeia (USP) 24 method. Tests were carried out in 900 mL of 50 mM sodium phosphate buffer (pH 6.8) containing 1% CTAB at 37 °C ± 0.5 (n = 3). All tests were done in triplicate using six tablets for each test.
Comment 5: Line 47 Is this a typing mistake? “the model was applied to model…”
We are thankful for the careful reviewing. Line 47 was revised.
the model was applied to the physical phenomena such as
Comment 6: Line 59, check if there is a typing mistake
Thank you for this comment, the revised sentence has been rewritten as follows:
our previous work that the simulation algorithms could simulate the disintegration and dissolution of the mefenamic acid (MA) tablets very close to real experimental data.
Comment 7: In lines 49 to 64, the authors presented some references about several modeling or simulation tools (FEM, DEM, Cellular-automata, molecular dynamics, etc.) which could be used for the elucidation of drug release from solid polymeric matrix. Nevertheless, the authors do not indicate which models will be using.
We are thankful for the careful review. Line65 to 66 was revised to indicate clearly that the use of cellular automata was used.
From the advantages of the three-dimensional cellular automata, the purpose of the study was to simulate the dissolution of the multi-layer tablets with the three-dimensional cellular automata,
Comment 8: Lines 82 to 91, it is not clear the method to tablet preparation and the tablet differences.
Thank you for the careful reviewing. The section of “2.2.1 preparation of tablets” was re-structured and revised.
Comment 9: Table 1 and 2, The weight for each tablet is not clear. In table 1, there was indicated that total tablet weight is 500 mg, but the authors have indicated that granulate weight was 980mg; the same in table 2. The tables must be explained in a better way
Thank you for the careful reviewing. The indicated weight for ASA layer was incorrect, so the value was revised.
Comment 10: Lines 128 to 133. Is this dissolution test method a USP monographic method? It looks is not, why the author have used this?
We are thankful for this careful review. The dissolution test method was based on the reference (Park, S.-H.; Choi, H.-K. The effects of surfactants on the dissolution profiles of poorly water-soluble acidic drugs. Int. J. Pharm. 2006, 321, 35–41.). However, the cone-effect was observed due to the large tablet size, so the paddle rotation speed was increased to 75rpm.
Comment 11: Lines 146 to 148, Why the authors used the same method of the molecular dynamics approach but in the introduction section, their focus was to FEM and DEM models?
Thank you for your reviewing. The molecular dynamics approach is to calculate the simulation constants used for the simulation of the drug release profiles with a cellular automata algorithm. To explain the purpose of using the molecular dynamics approach, Line 146 to 148 was revised.
In our previous work [30], the molecular dynamics approach was adopted to calculate the simulation constants of MA particle, which expresses the dissolution rate in the simulation model with a cellular automata, thus the same method of the molecular dynamics approach was adopted for the calculation of ASA simulation constants as well
Comment 12: Table 3, effective compressive stress calculation method is not indicated in materials and methods section
Thank you for the careful reviewing. The effective compressive stress calculation method was introduced in material and methods section 2.2.2.
Comment 13: Figure 2 and 3, error bars, legend and resolution of the figures must be improved. This figures are not clear and they do not help the understanding of the discussion. It is recommended the use of Graphpad prism software to improve the figures and the legends.
We are thankful for the comments. In order to help the readers and to improve visualization aspects, the figures were revised to improve the readability of the charts and chart elements.
Comment 14: Figures 5 and 6 are not clear. They have a lot of dots and other signs that cause confusion in the readers
The resolution of the figures was improved and other signs in Figures 4 (Figure 5 in the original manuscript) and 5 (Figure 6 in the original manuscript) were introduced to improve on the clarity of the data representation.
Comment 15: Lines 221 to 225; These lines lack of scientific discussion about the release profiles, the authors just indicated that there is a significant reduction in release, but they do not present statistical arguments for that. In addition, what about the diffusion process for the active principles in the aqueous media? this aspect could improve the discussion due to the inert layer provide a boundary condition for drug diffusion to medium. The used software could be used to analyze some parameters about the Fick´s Diffusion Law.
Thank you for providing these insights. In the statistical analysis of the dissolution rates at 10min, 15min and 30min between single entities, bilayer tablets and triple-layer tablets, significant difference (i.e. P<0.05) was only observed when comparing the dissolution between ASA tablets and triple-layer tablets. Therefore, the result of the statistical analysis was introduced into Line 221to 225.
i.e. in the statistical analysis (one way ANOVA) of dissolution rates at 10 min, 15 min and 30min, P-value<0.05 is only observed when comparing the dissolution rates between ASA tablets and triple-layer tablets
Comment 16: Lines 228 to 230. A detailed revision of limitation to drug diffusion and data analysis must be done. The erosion and swelling process for the inert layers must be addressed. It is recommended the following references to check:
Nikolett Kállai, Oliver Luhn, et.al. Evaluation of Drug Release From Coated Pellets Based on Isomalt, Sugar, and Microcrystalline Cellulose Inert Cores. AAPS PharmSciTech. 2010 Mar; 11(1): 383–391
Kállai-Szabó N, et.al. Comparative dissolution study of drug and inert isomalt based core material from layered pellets. J Pharm Biomed Anal. 2014 Sep;98:339-44. doi: 10.1016/j.jpba.2014.06.005. Epub 2014 Jun 10.
Wei HE, et.al. Design and in Vitro/in Vivo Evaluation of Multi-layer Film Coated Pellets for Omeprazole. Chem. Pharm. Bull. 57(2) 122—128 (2009).
Kyu-Mok Hwanga, et.al. Swellable and porous bilayer tablet for gastroretentive drug delivery: Preparation and in vitro-in vivo evaluation. International Journal of Pharmaceutics, Volume 572, 15 December 2019, 118783
Thank you for pointing out the relevant literature, we have introduced these references in the manuscript and addressed in the introduction the related aspects.
Comment 17: In lines 231 to 233. There is not a clear relationship between the geometry of the tablet and the ASA tablets disintegration rate. This section is lacking order and structure.
We are thankful for this constructive suggestion and comment. We have revised the mentioned sentences to address a tablet geometry influence on the disintegration and referenced one the above-suggested literature.
Comment 18: In lines 233 to 234
Comment 19: The authors talk about the difference in the experimental results, compared to the Noyes and Whitney model; however, the results are not adequately discussed, since the assumptions considered for the application of the Noyes-Withney model are not indicated
This comment is well accepted. The decision to use original Noyes-Whitney model as fitting function for experimental results, despite the mentioned limitations, was due to the fact that it includes physically relevant fitting parameters, such as diffusivity, thickness of an unstirred layer, solubility and surface. Our decision to use only surface as a fitting parameter for benchmarking tablets of the same formulations assumed that other parameters will not change. We have discussed some of the limitations in the Discussion section.
Comment 20: In lines 235 to 238 the authors indicated some aspects about the liquid imbibition; nevertheless, they did not any experiments where they addressed the liquid imbibition characterization, this evaluation does not appear in materials and methods section, it is just a speculation
We are thankful for this comment and for the careful review of the manuscript. Our assumption was not based on the actual experimental data but rather on the visual observation, which we have corrected in the text.
The disintegration mechanism of ASA layer is different to the one of MA layer and can be best characterized as being erosion-driven. The difference in the disintegration mechanisms for almost identical formulations, i.e., not taking into account the geometrical differences, can be best explained by different diffusivities and aqueous solubilities of the active substances. Interestingly, the visual observation of the tablets’ behavior during dissolution suggests rather limited liquid imbibition despite sufficient porosity; liquid transport in the porous meshwork is, apparently, not intensive enough to cause an immediate disintegration, therefore an erosion-driven disintegration occurs.
Comment 21: In lines 239 to 245, the authors gave a discussion about disintegration and it relationship with dissolution profiles, also they have implicated that the inclusion of calcium carbonate to the formulation could improve a fast disintegration but they do not conducted any experiment which supported these types of affirmations. The last discussion is not supported enough
We are thankful for the comments. The confusing statements were removed from the manuscript.
Comment 22: In the concluding section, the authors agree on the impact of the inert layer on the tablets. However, they had indicated that one of the objectives of the work was to in-silico modelling to simulate the drug dissolution profiles, even much of the introduction was on the FEM, DEM, cellular-automata simulation models, among others. However, the authors do not discuss or conclude on the simulation profiles and therefore the work lacks clear objectives and a logical and analytical order.
Thank you for providing the insights. To improve the consistency of the objectives and the conclusion in this manuscript, the sentence in the concluding section was revised.
The simulation model with the three-dimensional cellular automata algorithm showed the good agreements with the experimental dissolution data and predicted the impact of the insoluble layer on the dissolution of the APIs in both layers. Therefore, it is concluded that the proposed simulation approach is valuable to predict the influence of material attributes and process parameters on drug release from multicomponent and multiple-layer pharmaceutical tablets and help us develop the drug product formulation.
Reviewer 4 Report
In the work reported by Yokoyama et al, the effect of the separation layer on the disintegration and dissolution kinetics of multilayer tablets was studied. It is an interesting work although some minor points have to be completed by authors:
- On table 2, what is the meaning of B5 and B7 tablet’s parameters?
- Reference 24 is incomplete.
Author Response
Rebuttal letter
We are thankful for the time spend by the reviewers to help improve this manuscript. All comments are well taken. Our answers to the original questions are marked in blue.
Reviewer-4 Comments
In the work reported by Yokoyama et al, the effect of the separation layer on the disintegration and dissolution kinetics of multilayer tablets was studied. It is an interesting work although some minor points have to be completed by authors:
- On table 2, what is the meaning of B5 and B7 tablet’s parameters?
Thank you for this comment, the misleading column was removed from the Table 2.
- Reference 24 is incomplete.
Thank you very much for pointing out to this typographical error. The reference 32 (reference 24 in the original manuscript) has been corrected.
Round 2
Reviewer 2 Report
The authors responded to my earlier decision of REJECT, some of the answers are satisfactory. However, the revised introduction does not actually point out the innovation of the article and the difference with ref. 30. The discussion of drug dissolution and diffusion is too simple and must be experimentally demonstrated rather than documented in introduction. Noyes-Whitney dissolution model discussion is still too simple. Therefore, I stand on my first decision.
Author Response
Reviewer-2 Comments
The authors responded to my earlier decision of REJECT, some of the answers are satisfactory. However, the revised introduction does not actually point out the innovation of the article and the difference with ref. 30. The discussion of drug dissolution and diffusion is too simple and must be experimentally demonstrated rather than documented in introduction. Noyes-Whitney dissolution model discussion is still too simple. Therefore, I stand on my first decision.
We understand the criticism of the reviewer concerning the use and application of the Noyes-Whitney model for description of the complex formulations, such as multiple-layer tablets. In fact, we absolutely agree with this statement, that the simplicity of this equation is not allowing it to be applied for description of the heterogeneous media, such as pharmaceutical tablets, unless those tablets are single component formulations. Therefore, we have applied this model only as a fitting function for comparative analysis of the obtained experimental data and as an oversimplified alternative method to our main modelling approach based on cellular automata, as was shown in the ref. 36 (former ref.30). The corresponding text passages were added between lines 201-205.
The entire complexity of the studied formulations could only be mathematically modelled by applying numerical methods, such as the one used in this manuscript and in ref. 36, i.e., the three-dimensional cellular automata. From a computational point of view the cellular automata are computationally inexpensive approximations for solutions of the parabolic differential equations, which are classical models for diffusion and liquid sorption. We have investigated the applicability of the numerical approach to model the dissolution profiles of the mefenamic acid tablets in our previous submission. The results of the cellular automata simulations of the acetylsalicylic acid and mefenamic acid tablets show particularly good agreement with experimental data. These results allow us to propose an explanation of the phenomenon, which was not intuitively expected, namely the difference in release of acetylsalicylic acid from binary and triple layer tablets, provided that the layers’ geometry was identical.
In order to support an assumption that the due to greater aqueous solubility of ASA as compared to MA, the water uptake, and therefore, disintegration and dissolution have little or no influence by MA layer, the liquid uptake experiment were carried out. The obtained results corroborate the previously taken assumption. The methods of these measurements were introduced in lines 164-174, section 2.2.5. The results and discussion were introduced in lines 241-248 and lines 288-290.
In the reference 36, we have concentrated our efforts on the validation of our modelling approach with a formulation of mefenamic acid, whereas in this study an application of numerical simulation to elucidate the root cause of the release difference was carried out.
In order to help the reader to better differentiate between generalized and discrete modeling approaches, we have extended the Discussion section between lines 310-326 and have added new references to the manuscript (lines 51-54).

Reviewer 3 Report
Now, it was easier to track the changes made in the manuscript. Also, this has improved considerably and therefore, I agree for its publication in the current form.
Author Response
Reviewer-3 comments
Now, it was easier to track the changes made in the manuscript. Also, this has improved considerably and therefore, I agree for its publication in the current form.
We are thankful for time and effort spend by reviewer to help us improve the manuscript.